# Modulation of Epithelial–Mesenchymal Transition Is a Possible Underlying Mechanism for Inducing Chemoresistance in MIA PaCa-2 Cells against Gemcitabine and Paclitaxel

**DOI:** 10.3390/biomedicines12051011

**Published:** 2024-05-03

**Authors:** Hajime Nakamura, Megumi Watanabe, Kohichi Takada, Tatsuya Sato, Fumihito Hikage, Araya Umetsu, Joji Muramatsu, Masato Furuhashi, Hiroshi Ohguro

**Affiliations:** 1Departments of Medical Oncology, School of Medicine, Sapporo Medical University, S1W17, Chuo-ku, Sapporo 060-8556, Japan; nakamurahajime27@sapmed.ac.jp (H.N.); ktakada@sapmed.ac.jp (K.T.); sapsupsap@gmail.com (J.M.); 2Departments of Ophthalmology, School of Medicine, Sapporo Medical University, S1W17, Chuo-ku, Sapporo 060-8556, Japan; watanabe@sapmed.ac.jp (M.W.); fuhika@gmail.com (F.H.); araya.umetsu@sapmed.ac.jp (A.U.); 3Departments of Cardiovascular, Renal and Metabolic Medicine, Sapporo Medical University, S1W17, Chuo-ku, Sapporo 060-8556, Japan; satatsu.bear@gmail.com (T.S.); furuhasi@sapmed.ac.jp (M.F.); 4Departments of Cellular Physiology and Signal Transduction, Sapporo Medical University, S1W17, Chuo-ku, Sapporo 060-8556, Japan

**Keywords:** 3D spheroid culture, pancreatic ductal carcinoma, RNA sequencing, ingenuity pathway analysis (IPA), gemcitabine, paclitaxel, Seahorse cellular metabolic analysis

## Abstract

To elucidate the currently unknown molecular mechanisms responsible for the similarity and difference during the acquirement of resistance against gemcitabine (GEM) and paclitaxel (PTX) in patients with pancreatic carcinoma, we examined two-dimensional (2D) and three-dimensional (3D) cultures of parent MIA PaCa-2 cells (MIA PaCa-2-PA) and their GEM resistance cell line (MIA PaCa-2-GR) and PTX resistance (MIA PaCa-2-PR). Using these cells, we examined 3D spheroid configurations and cellular metabolism, including mitochondrial and glycolytic functions, with a Seahorse bio-analyzer and RNA sequencing analysis. Compared to the MIA PaCa-2-PA, (1) the formation of the 3D spheroids of MIA PaCa-2-GR or -PR was much slower, and (2) their mitochondrial and glycolytic functions were greatly modulated in MIA PaCa-2-GR or -PR, and such metabolic changes were also different between their 2D and 3D culture conditions. RNA sequencing and bioinformatic analyses of the differentially expressed genes (DEGs) using an ingenuity pathway analysis (IPA) suggested that various modulatory factors related to epithelial –mesenchymal transition (EMT) including STAT3, GLI1, ZNF367, NKX3-2, ZIC2, IFIT2, HEY1 and FBLX, may be the possible upstream regulators and/or causal network master regulators responsible for the acquirement of drug resistance in MIA PaCa-2-GR and -PR. In addition, among the prominently altered DEGs (Log2 fold changes more than 6 or less than −6), FABP5, IQSEC3, and GASK1B were identified as unique genes associated with their antisense RNA or pseudogenes, and among these, FABP5 and GASK1B are known to function as modulators of cancerous EMT. Therefore, the observations reported herein suggest that modulations of cancerous EMT may be key molecular mechanisms that are responsible for inducing chemoresistance against GEM or PTX in MIA PaCa-2 cells.

## 1. Introduction

It is known that pancreatic ductal adenocarcinoma (PDAC) is a malignant tumor that has a poor prognosis, and in fact, even after undergoing potentially curative surgery, their 5-year survival rate is only approximately 15–25% [1,2,3]. In terms of the clinical treatment for most patients with PDAC, systemic chemotherapy is conducted regardless of the surgical treatment option. In the past decade, based upon evidence obtained from various clinical trials [4,5,6,7,8], a combination of gemcitabine (GEM) and an albumin nanoparticle conjugate of paclitaxel (nab-paclitaxel, n-PTX) [6,9] have emerged as a first-line therapy in patients with advanced PDAC. However, the response to this chemotherapy regimen is still poor because of the rapid acquirement of drug resistance in most patients [10]. Therefore, identifying the underlying molecular mechanisms responsible for causing such chemoresistance to be acquired, as well as additional candidate targets and compounds that can overcome these factors, are urgently required. However, for this purpose, the various in vitro drug screening methods using conventional two-dimensional (2D) planar cultures of cancer cell lines have been used, but using these cultures, it has not been successfully identified possible candidate drugs and compounds to translate into clinical applications [11,12,13,14]. In addition, recent studies have suggested that the molecular mechanisms responsible for inducing this chemoresistance are much more complicated because of the numerous genetic changes that are related to various cellular signaling pathways and responses of PDAC cells [15], in addition to drug transport [16] and the tumor microenvironment [17]. Therefore, developing a better understanding of those underlying mechanisms would make it possible to identify promising therapeutic strategies for overcoming this chemoresistance. To accomplish this, it will be necessary to develop more suitable in vitro models that replicate the biological characteristics of the PDAC tumor environment.

Three-dimensional (3D) cultures were developed in order to replicate in vitro tumor models more closely [18,19]. Among the various types of in vitro 3D cell culture models, an in vitro 3D spheroid model is the simplest and, thus, has been the most frequently used in studies related to not only cancerous but also non-cancerous related research fields [20,21]. In fact, 3D spheroid cultures of various PDAC cell lines are now recognized as a better in vitro model to mimic the tumor microenvironment in investigations related to tumor pathophysiology and chemoresistance, as well as in drug screening [22,23,24]. Our group recently independently developed various in vitro 3D spheroid models using various non-cancerous ocular-related cells [25,26,27,28] and rat cardiomyocytes, H9c2 cells [29], as well as cancerous cell lines including an A549 lung adenocarcinoma cell line [30], malignant melanoma cell lines [31] and oral squamous cell carcinoma (OSCC) cells [32]. These collective studies allowed us to conclude that the biological natures were significantly different between the 2D planar cultures and 3D spheroid cultures, even though we employed exactly the same experimental conditions except that different culture plates were used. Interestingly, we also found that the appearance of the 3D spheroids was also different between non-cancerous and cancerous cells, in that they were globe-shape [26,27,28] or non-globe shape [30,31,32], respectively. Furthermore, we also recognized that the appearances of the 3D spheroids were significantly diverse among malignant tumors even though they had the same origins, and the degree of difference was potentially correlated with cellular metabolic functions, pathological aspects, and/or cytotoxicity against anti-tumor drugs [30,31,32]. Considering these collective findings, we concluded that characteristic appearances of the cancerous 3D spheroid could be a potential indicator for evaluating the clinicopathological aspects of certain malignant tumors.

Therefore, in the current study, to elucidate the currently unidentified underlying molecular mechanisms responsible for the chemoresistance of PDAC against GEM or PTX, using a well-characterized PDCA cell line, MIA PaCa-2 as a parent cell line (MIA PaCa-2-PA), their corresponding chemoresistant cell lines against GEM (MIA PaCa-2-GR) or PTX (MIA PaCa-2-PR) were prepared. Thereafter, those were further cultured by 2D planar and 3D spheroid culture methods, and the resulting cultures were then subjected to a Seahorse real-time cellular metabolic analysis. In addition, three 2D cultured cell lines were also subjected to RNA sequencing analysis in an attempt to elucidate possible critical genes responsible for developing chemoresistance against GEM or PTX in PDAC.

## 2. Materials and Methods

The current study, which was conducted at the Sapporo Medical University Hospital, Japan, was approved by the institutional review board (IRB, registration number 342-3416) according to the tenets of the Declaration of Helsinki and national laws for using human-related carcinoma cell lines.

### 2.1. Preparations of Gemcitabine or Paclitaxel Resistance MIA PaCa-2 Cells

A pancreatic ductal carcinoma cell line, MIA PaCa-2 cells, was obtained from the American Type Culture Collection (Manassas, VA, USA). MIA PaCa-2 cells were cultured in 2D culture dishes at 37 °C in a 2D culture medium composed of HG-DMEM culture medium supplemented with 8 mg/L d-biotin, 4 mg/L calcium pantothenate, 100 U/mL penicillin, 100 μg/mL streptomycin (b.p. HG-DMEM), 10% CS were used as the parental line (MIA PaCa-2-PA). For generating drug resistance in MIA-PaCa-2 against GEM (MIA PaCa-2-GR) or PTX (MIA PaCa-2-PR), subcultures of the MIA PaCa-2-PA cells were exposed to incremental increases in GEM or PTX concentrations, starting with an IC50 dose (GEM: 26 nM, PTX: 232 nM) for six months. Finally, the MIA PaCa-2-GR or MIA PaCa-2-PR cells developed the capacity for proliferation when returned to a medium containing 2.6 µM GEM or 5.0 µM PTX, respectively.

### 2.2. 3D Cell Cultures of MIA PaCa-2-P, -GR or -PR Cells

MIA PaCa-2-PA, -GR, or -PR cells, as generated above, were 3D cells cultured by methods described in previous reports using 3T3-L1 preadipocytes and human orbital fibroblasts [26,33,34,35,36,37]. Briefly, these cells were each cultured in 2D culture dishes at 37 °C in a 2D culture medium containing 0.25% *w*/*v* Methocel A4M in the absence or presence of 2.6 μM GEM or 0.5 μM PTX until reaching approximately 90% confluence. After washing with a phosphate-buffered saline (PBS), the cells were detached by treatment with 0.25% Trypsin/EDTA, resuspended in the culture medium, and 28 μL of medium containing approximately 20,000 cells were added to each well of the drop culture plate (# HDP1385, Sigma-Aldrich, Burlington, MA, USA) (3D/Day 0) as described previously [26,34]. Thereafter, half of the culture medium was replaced with fresh medium in each well daily until Day 5 [35,36,37]. As a representative non-cancerous human cell line, human trabecular meshwork (HTM) cells (Applied Biological Materials Inc., Richmond, BC, Canada) [38] were also used. The 3D spheroid morphology was observed by a phase contrast microscope (Nikon ECLIPSE TS2; Tokyo, Japan), as described previously [35,36,37].

### 2.3. Real-Time Analysis of the Cellular Metabolic Functions 

Oxygen consumption rate (OCR) and extracellular acidification rate (ECAR) in 2D and 3D cultured MIA PaCa-2-P, -GR, or -PR cells were measured using a Seahorse XFe96 real-time metabolic analyzer (Agilent Technologies, Santa Clara, CA, USA). On the day of assay, an XFe96 Cell Culture Microplate (Agilent Technologies, #103794-100) was coated with Cell-Tak™ (Corning #354240, Corning, NY, USA). In brief, 200 μL of 2 mg/mL Cell-Tak in 5% acetic acid was added in 2.8 mL of 0.1 M sodium bicarbonate, and then 30 μL of this Cell-Tak Mix was placed in each well of a microplate and incubated for 1 h in a non-CO_2_ incubator at 37 °C. Following the incubation, Cell-Tak Mix was aspirated from the plate, and the plate was washed twice with 400 µL of sterile 37 °C water and allowed to air dry. Approximately 10,000 2D-cultured cells and six 3D-cultured spheroids were resuspended to a pre-warmed 50 μL Seahorse XF DMEM assay medium (pH 7.4, Agilent Technologies, #103575-100) containing 5.5 mM glucose, 2.0 mM glutamine, and 1.0 mM sodium pyruvate and were seeded onto each well in the pre-made Cell-Tak coated Seahorse assay plate. The plate was incubated in a CO_2_-free incubator at 37 °C for 1 h prior to the measurements.

OCR and ECAR were measured in an XFe96 extracellular flux analyzer at the baseline and after the following sequential injections of 2.0 μM oligomycin, 5.0 μM

carbonyl cyanide-*p*-trifluoromethoxyphenylhydrazone (FCCP), a mixture of 1.0 μM rotenone, 1.0 μM antimycin A, and 10 mM 2-deoxyglucose (2DG). The OCR and ECAR values were normalized for the number of protein contents assessed by a BCA protein assay (TaKaRa, Otsu, Japan) per well by lysing the cells of the wells in which the measurements were completed with 10 μL of CelLytic™ MT Cell Lysis Reagent (Sigma-Aldrich). Key parameters of mitochondrial and glycolytic functions were calculated as follows: Basal respiration = OCR at baseline − OCR after adding R/A; ATP-linked respiration = OCR at baseline − OCR after adding oligomycin; Maximal respiration = OCR after adding FCCP − OCR after adding R/A; Glycolytic capacity = ECAR after adding oligomycin − ECAR at baseline; Glycolytic reserve = ECAR after adding oligomycin − ECAR after adding 2DG.

### 2.4. RNA Sequencing, Gene Function, and Analysis of Pathways

Total RNA was isolated from 2D confluent cells of MIA PaCa-2-PA, -GR, or -PR in a 150 mm dish as described above (*n* = 3) using an RNeasy mini kit (Qiagen, Valencia, CA, USA) according to the manufacturer’s instructions and were then subjected to an RNA sequencing analysis as described recently [30]. Briefly, after the RNA content and quality were checked to make sure that the RNA quality was suitable for RNA sequencing, ribosomal RNA was removed from total RNA using NEBNext^®^ Poly(A) mRNA Magnetic Isolation Module (Cat. # E7490, New England BioLabs, Ipswich, MA, USA). The rRNA-depleted RNA was then processed to convert to cDNA using a TruSeq RNA Sample Preparation Kit (Illumina, San Diego, CA, USA) and final sequence-ready libraries with the NEBNext Ultra II RNA library prep kit (Cat. #E7760, New England BioLabs). After checking their quality and quantity using an Agilent 2100 Bioanalyzer and KAPA Library Quantification Kit (KAPA Biosystems, Wilmington, MA, USA), respectively, they were subjected to NovaSeq 6000 and GenoLab M sequencing in the PE150 mode. Sequence data were filtered by removing the adapter sequence, ambiguous nucleotides, and low-quality sequences using software (FastQC, version 0.11.7) as quality control by an Agilent 2100 Bioanalyzer (Agilent, CA, USA) and Trimmomatic (version 0.38) were mapped to the reference genome sequence (GRCh38) using HISAT2 tools software [39]. The read counts for each respective gene and statistical analysis were analyzed by featureCounts (version 1.6.3) and DESeq2 (version 1.24.0), respectively. Statistical significance was determined by an empirical analysis, and genes with fold-change ≥ 2.0 and FDR-adjusted *p*-value < 0.05 and *q* < 0.08 were assigned as differentially expressed genes (DEG).

To predict possible upstream transcriptional regulators, DEGs were interpreted using the upstream regulator and causal network regulator functions of the ingenuity pathway analysis (IPA, Qiagen, accessed on 27 July 2023. https://digitalinsights.qiagen.com/products-overview/discovery-insights-portfolio/analysis-and-visualization/qiagen-ipa/) [40]. 

### 2.5. Other Analytical Methods

For drug sensitivity measurements, cells were seeded in 96-well plates at a density of 3 × 10^3^ cells/well and allowed to attach for 24 h, after which they were cultured for 72 hrs with 0–100 μM GEM or PTX. Cell viability was evaluated by a WST-1 assay (Premix WST-1 Cell Proliferation Assay; Takara Bio, Otsu, Japan) and Infinite M1000 PRO microplate reader (Tecan Japan, Kawasaki, Japan). Absorbance was measured at 450 nm to determine cell viability.

Quantitative PCR using specific primers (Appendix A) was conducted with Quant Studio 3 (Applied Biosystems, Foster City, CA, USA). RNA was extracted using TRIzol Reagent (Thermo Fisher Scientific, Waltham, MA, USA) according to the manufacturer’s protocol, and 1 µg of total RNA was reverse transcribed with a SuperScript VILO cDNA synthesis kit (Thermo Fisher Scientific). The analysis was conducted in quadruplicate using a POWER UP SYBR Green Master Mix (Thermo Fisher Scientific). Transcript levels were normalized to β-actin expression. 

Statistical analyses using the Graph Pad Prism 8 (GraphPad Software, San Diego, CA, USA) were performed as demonstrated in a previous report [35]. All statistical analyses were performed using the Graph Pad Prism 8 (GraphPad Software, San Diego, CA). The statistical difference between groups was determined using a Students’ *t*-test for two-group comparison, or two-ANOVA followed by a Tukey’s multiple comparison test. Data are expressed as the arithmetic mean ± the standard error of the mean (SEM).

## 3. Results

To elucidate biological similarities and differences upon acquiring chemoresistance of PDCA against various anti-cancer drugs, a well-characterized cell line, MIA PaCa-2, was used. Initially, using MIA PaCa-2-PA and standard first-line anti-tumor drugs, GEM or PTX, MIA PaCa-2-GR, and -PR were prepared. As shown in Figure 1, chemoresistance against GEM or PTX were apparently obtained in the MIA PaCa-2-GR (GEM IC 50 = 138 nM) or MIA PaCa-2-PR (PTX IC50 = 1052 nM), while this was not the case for MIA PaCa-2-PA (GEM IC50 = 26 nM, PTX IC50 = 232 nM), respectively.

In our recent study, we reported that the 3D spheroid configurations were significantly diverse among malignant tumor cells even though they had the same origin, and these biological diversities were confirmed by Seahorse cellular metabolic measurements [31]. Therefore, we concluded that 3D spheroid cultures of malignant tumors might be quite useful for evaluating diverse biological aspects among various malignant tumors. In the current investigation, using this methodology, unidentified biological similarities and differences among three cell lines, MIA PaCa-2-PA, -GR, and -PR, were studied. As shown in Figure 2, the cells that were placed into each well of the 3D drop culture all began to coalesce after one day, as was typically observed in the 3D spheroid cultures of none cancerous and cancerous cells [26,27,28,30,31,41], but the sizes were apparently larger in MIA PaCa-2 cell lines. However, unexpectedly, the progression for forming 3D spheroids in the MIA PaCa-2-PA was extremely slow and did not progress to the formation of a solid 3D spheroid as is typically recognized within the non-cancerous and cancerous cells [26,27,28,30,31,41] after 5 days of culture. Such slower 3D spheroid formation was more evident in the chemoresistant cell lines, MIA PaCa-2-GR, and -PR. 

We then studied the cellular metabolic characteristics of the 2D and 3D cultured three cell lines, MIA PaCa-2-PA, -GR, and -PR. As shown in Figure 3, as compared with MIA PaCa-2-PA, the mitochondrial (OCR) and glycolytic functions (ECAR) of MIA PaCa-2-GR and -PR were significantly modulated, and those changes were also different between 2D and 3D cultures. That is, (1) in the 2D culture, both OCR and ECAR indices were substantially increased in the order of MIA PaCa-2-PR and -GR, and (2) in the 3D cell cultures, ECAR indices were similarly modulated as the 2D cultured cells, but within the OCR indices, basal respiration and ATP-linked respiration were markedly decreased in the order of MIA PaCa-2-GR and -PR. Therefore, these collective observations indicate that (1) biological aspects are greatly modulated on acquiring chemoresistance, and these aspects were also different between GEM and PTX, and (2) even though solid 3D spheroids were not generated, significant alterations of the biological functions of MIA PaCa-2 related cells were induced in the 3D cultures, as compared with the corresponding 2D planar cell cultures. 

RNA sequencing analyses were performed in an attempt to elucidate the currently unidentified mechanisms responsible for inducing such characteristic biological alterations upon acquiring chemoresistance against GEM or PTX in the MIA PaCa-2 cells. As shown in the heatmap (Figure 4) and MA and volcano plots (Appendix A), 578, 991, or 1319 significantly up-regulated and 890, 873 or 800 down-regulated differentially expressed genes (DEGs) were identified between MIA PaCa2-PA vs. -GR, MIA PaCa-2-PA vs. -PR or MIA PaCa2-GR vs. -PR, respectively, with a significance level of <0.05 (FDR) and an absolute fold-change ≥2 was identified (the list of all of the up-regulated and down-regulated DEGs is attached in a Appendix A). Among these DEGs, the most prominently up-regulated or down-regulated DEGs (Log2 fold change of more than 6 or less than −6, respectively) were compared between MIA PaCa-2-PA vs. -GR and MIA PaCa-2-PA vs. -PR. As shown in Table 1 and Table 2, a total of three up-regulated DEGs (DACH1, VCAM1, and PACRG) and 11 down-regulated DEGs (TGFBR2, DOC2B, VCAN, MDFIC, LRP1B, GLIS3, MMP1, DSC2, ARHGDIB, ARGGAP15 and ZNF488) were commonly detected. We, therefore, speculated that these genes are most likely involved in the molecular mechanisms responsible for inducing chemoresistance of MIA PaCa-2 cells regardless of different anti-tumor drugs. Alternatively, a pair of FABP5 and FABP5P7, IQSEC3 and IQSEC3-AS1, and GASK1B and GASK1B-AS1 were identified only in the MIA PaCa-2-PA vs. -GR or MIA PaCa-2-PA vs. -PR, respectively. We speculate that these unique DEGs that were detected in one of both chemoresistant MIA PaCa-2 cells may be key regulatory factors for acquiring chemoresistance against GEM or PTX.

To examine these issues further, we conducted an Ingenuity Pathway Analysis (IPA, Qiagen, Redwood City, CA) to estimate possible up-stream regulators and causal network master regulators between MIA PaCa-2-PA vs. -GR (PA vs. GR), MIA PaCa-2-PA vs. -2 PR (PA vs. PR) or MIA PaCa-2-GR vs. -PR (GR vs. PR), respectively. As shown in Table 3, the results indicated that 2 or 4 (PA vs. GR), 3 or 8 (PA vs. PR), and 3 or 9 (GR vs. PR) candidate genes were estimated as possible upstream regulators or causal network masters, respectively. It was speculated that the observed DEGs and their related estimations should be included in two possible mechanisms, that is, (1) biological deteriorations and/or cellular damage caused by GEM or PTX, and (2) newly acquired biological activities for survival in the presence of GEM or PTX. Taking into account the possible roles of cancer progression of each up-stream regulator and the causal network master regulators, which have already been elucidated (Table 3), we rationally speculate that IFIT2-related signaling or STAT3, GLI1, ZNF367, NKX3-2, ZIC2, HEY1, TAP1, and FBXL14 related signaling represent possible candidates involved in the underlying molecular mechanisms causing chemoresistance against GEM or PTX, respectively, in addition to three possible factors, i.e., FABP5, IQSEC3, and GASK1B as above. Among these candidate genes, the qPCR analysis (Figure 5) confirmed that IFIT2 and FABP5 or STAT3, GLI1, NKX3-2, ZIC2, HEY1, TAP1, and GASK1B may be truly possible master regulators for inducing GEM or PTX resistance in MIA PaCa-2 cells.

## 4. Discussion

It is known that GEM, n-PTX, and other anti-cancer drugs are effective in the treatment of patients with advanced and metastatic PDAC, but acquiring chemoresistance to these drugs seriously deteriorates their effectiveness. However, although those underlying molecular mechanisms have not yet been fully identified, various transcription factors and signaling pathways involved in nucleoside metabolism are possible candidates for being involved in the development of such chemoresistance [61,62,63,64]. Theoretically, possible underlying molecular mechanisms for causing acquired chemoresistance include drug transport, drug-induced effects on various enzymes and others. It is likely that drug transport, activation, and metabolism are precisely regulated by numerous enzymes, and therefore, acquiring chemoresistance is thought to be regulated by various additional factors, including the tumor microenvironment, EMT, microRNA, and others [64]. Among these mechanisms, it is well known that the EMT phase converts phenotypes into tumor cells in which aggressive EMT changes are evoked and thus associated with their morphological changes as well as various alterations in genome and protein levels. Alternatively, it has also been reported that such mesenchymal transcription factors are pivotal factors in the induction of chemoresistance [64]. In fact, in addition to two key transcription factors, Snail and Twist, various signaling pathways could also be responsible factors such as Notch, tumor necrosis factor-alpha (TNFα), transforming growth factor beta (TGF-β), and hypoxia-inducible factor-1 alpha (HIF1α), which are involved in the induction of EMT in pancreatic cancer cells [65]. Quite interestingly, in the current study, all of the up-stream regulators and/or causal network master regulators estimated by the IPA analysis of the RNA sequencing, that is, STAT3 [42], GLI1 [43], ZNF367 [53], NKX3-2 [47], ZIC2 [51] IFIT2 [46], HEY1 [55] and FBXL14 [60] were identified as being directly or indirectly related to the EMT mechanisms of cancerous cells. Furthermore, among four factors, FABP5, IQSEC3, GASK1B, and SCN1A were identified as unique genes associated with their antisense RNA or pseudogene among the prominently altered DEGs (Log2 fold changes more than 6 or less than −6), FABP5 [66] and GASK1B [67] are also identified as modulators of cancerous EMT. Alternatively, despite the lack of evidence of any apparent correlation with cancerous EMT, IQSEC3 was identified as a novel prognostic marker for breast cancer patients [68], and it is known that SCN1A is a possible factor in the development of chemoresistance in esophageal adenocarcinoma [69]. Furthermore, VCAM1 was identified as the top 7 and the top 2 significant up-regulated DEG in MIA-GR and MIA-PR, respectively (Table 1). In fact, VCAM1 has been suggested as a factor in estimating poor patient prognosis and can promote tumor metastasis by inducing EMT in cancer [70]. In pancreatic cancer, EMT is known to lead to acquiring the characteristics of cancer stemness and enhance chemotherapy resistance through multiple different ways [71]. For instance, it is reported that the recapitulation of the fibrotic rigidities in pancreatic cancer tissues promotes elements of EMT, and the stiffness induces chemoresistance in pancreatic cancer cells [72]. Based on these findings, we suggested that VCAM1 is a potential regulator of the acquired resistance to GEM and n-PTX, which needs further investigation in the future. Therefore, these collective observations suggest that modulations of the cancerous EMT phenotype may be the main factor in the underlying molecular mechanisms for the induction of chemoresistance against GEM or PTX in MIA PaCa-2 cells.

Recent studies related to the field of cancer biology have pointed to the biological importance of the tumor surrounding environment (TSE) in addition to cancerous cells themselves because of their great influence on tumorigenesis, progression, metastasis, and drug sensitivities [73,74,75,76]. To study this further, in contrast to the conventional in vitro 2D planar cell culture models, in vitro 3D cell culture models will be required because of their high potential for replicating the physiological and spatial local environments of cancerous cells. In fact, such 3D cell culture methods are being more frequently applied for testing not only concerning drug efficacy but also in determining suitable dosages for chemotherapy [77,78,79,80,81]. Among the numerous in vitro 3D cell culture methods [20,21], we successfully produced various simple in vitro 3D spheroids using non-cancerous cells [26,27,28] as well as cancer cells [30,31] and found that the physical properties of the 3D spheroid represent potentially new indicators for estimating the biological nature of cancerous cells such as malignancy and drug efficacies [30,31,32,41]. In the current investigation, unfortunately, we were only successful in producing pre-matured 3D spheroids but not solid 3D spheroids, which are usually generated from most non-cancerous [26,27,28] and cancerous cells [30,31] using MIA PaCa-2 cells. However, similar to our results, 3D spheroid cultures of PDAC-related cells such as PANC-1 and MIA PaCa-2 were extremely difficult to produce, and in fact, much longer culture periods in excess of 10 days were required to obtain quite soft 3D spheroids [82,83,84]. Alternatively, to produce more solid 3D spheroids, co-culturing with other cells, such as cancer-associated fibroblasts (CAF), is required in addition to PDAC [84,85]. However, in the current study, the mitochondrial and glycolytic functions of the 3D MIA PaCa-2 spheroids were significantly different from those of their 2D cultured cells, and these differences in the cellular metabolic functions between 2D and 3D cultured cells closely resembled 3T3-L1 cells [33]. Therefore, even though such soft and pre-mature 3D spheroids were formed, their biological natures may already have been altered as the solid 3D spheroid. Collectively, 3D spatial environments rather than solid formation may be required for developing a complete understanding of the biological significance of the in vitro 3D spheroid models.

The present study showed that metabolic capacities were significantly increased in MIA PaCa-2-GR and MIA PaCa-2-PR compared to those in MIA PaCa-2-PA. Such increased metabolic capacities were observed in both 2D- and 3D-culture conditions. Activation in metabolic pathways has been reported as one phenotype of metabolic plasticity, a finding that can be observed in cancer cells that are resistant to chemotherapy [86], which is consistent with the findings in the present study. The molecular mechanisms underlying chemotherapy resistance-induced metabolic alteration are presumably multifactorial, but the most plausible interpretation would be a compensatory response of cancer cells to secure energy for survival against anti-cancer agents. Indeed, it has also been reported that EMT can activate both oxidative phosphorylation and glycolysis [87]. Interestingly, the degree of increased metabolic capacity in MIA-PaCa-2-GR was milder than that in MIA-PaCa-2-PR. The finding that the gene expression level of FABP5, an important lipid chaperon for the activation of intracellular metabolism in cancer cells [88,89], was markedly different between MIA-PaCa-2-GR and MIA-PaCa-2-PR might be one of the explanations for the differences in metabolic capacities between two cells. Nevertheless, an enhanced metabolic capacity in chemotherapy-resistant cells may play a significant role in cell survival and function. The assessment of cellular metabolism in the acquisition of chemotherapy resistance may be an important factor in the selection of anti-cancer therapies.

In conclusion, our current observations using RNA sequencings suggest that modulations of cancerous EMT may be key underlying molecular mechanisms that are responsible for inducing chemoresistance against GEM or PTX in MIA PaCa-2 cells. However, the 3D spheroid appearance and cellular metabolic aspects of the GEM or PTX-resistant MIA PaCa-2 were significantly different from each other, and possible upstream and causal network master regulators inducing their chemoresistance estimated by IPA analysis were also different. Therefore, these collective observations suggest that the acquirement of chemoresistance against GEM or PTX may associate diverse modulations of the cancerous EMT, and thus, additional investigation to elucidate these unidentified issues will be required as our next project. 

## Figures and Tables

**Figure 1 biomedicines-12-01011-f001:**
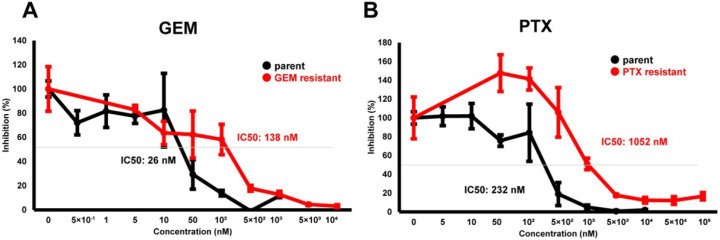
Cytotoxic analysis of chemoresistant MIA PaCa-2 cells. To determine the cytotoxicity against GEM (**A**) or PTX (**B**) in MIA PaCa-2-PA, MIA PaCa-2-GR, and MIA PaCa-2-PR, survival living cells detected using a WST-1 assay were plotted (*n* = 3).

**Figure 2 biomedicines-12-01011-f002:**
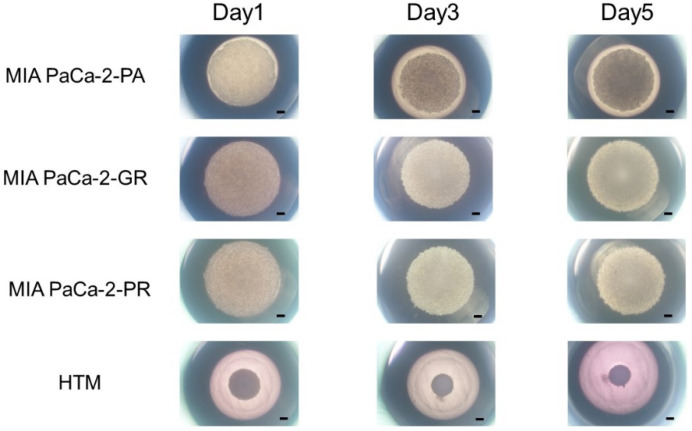
3D spheroid cultures of MIA PaCA-2-PA and -GR or -PR. Approximately 20,000 cells of MIA PaCa-2-PA, MIA PaCa-2-GR, or MIA PaCa-2-PR were subjected to a 3D drop cell culture to form 3D spheroids. As a representative non-cancerous human cell line, 20,000 cells of human trabecular meshwork (HTM) were also subjected to the 3D spheroid culture. Representative phase contrast microscopy images of these cells on Day 1, 3, and 5. Scale Bar: 100 μm.

**Figure 3 biomedicines-12-01011-f003:**
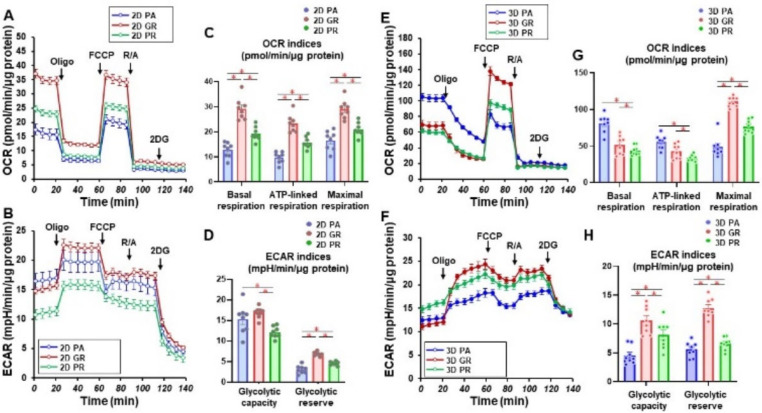
Measurement of mitochondrial and glycolytic functions of MIA PaCa-2-PA and -GR or -PR. Real-time metabolic function analysis by an XFe96 Extracellular Flux Analyzer of the 2D- or 3D-cultured MIA PaCa-2 cells in fresh preparations (*n* = 8). (**A**) Measurement of OCR in 2D-cultured cells. (**B**) Measurement of ECAR in 2D-cultured cells. (**C**) Key parameters in mitochondrial function in 2D-cultured cells. (**D**) Key parameters in glycolytic function in 2D-cultured cells. (**E**) Measurement of OCR in 3D-cultured spheroids. (**F**) Measurement of ECAR in 3D-cultured spheroids. (**G**) Key parameters in the mitochondrial function in 3D-cultured spheroids. (**H**) Key parameters in glycolytic function in 3D-cultured spheroids. OCR, oxygen consumption rate; ECAR, extracellular acidification rate; Oligo, oligomycin; FCCP, carbonyl cyanide *p*-trifluoromethoxyphenylhydrazone; R/A, otenone/antimycin A; 2DG, 2-deoxyglucose. * *p* < 0.05 (one-way ANOVA followed by a Tukey’s muliple comparison test).

**Figure 4 biomedicines-12-01011-f004:**
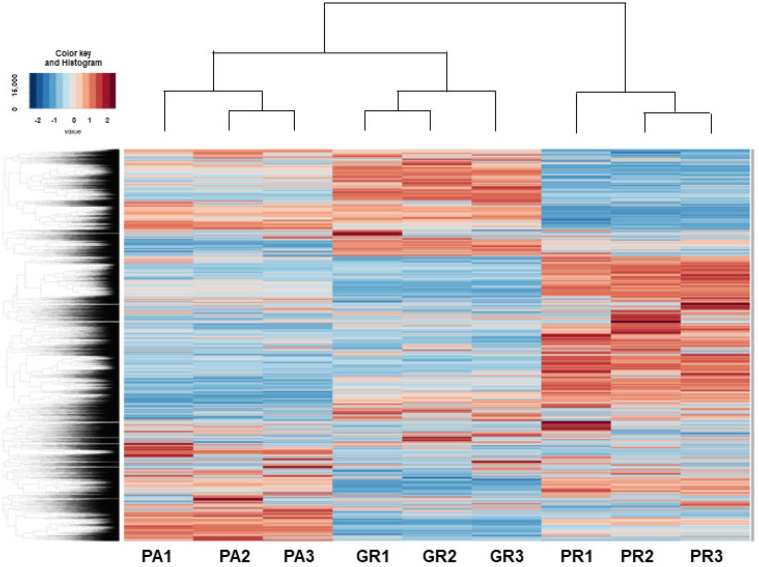
Heatmap for DEGs between 2D cultured MIA PaCA-2-PA and -2-GR or -PR.

**Figure 5 biomedicines-12-01011-f005:**
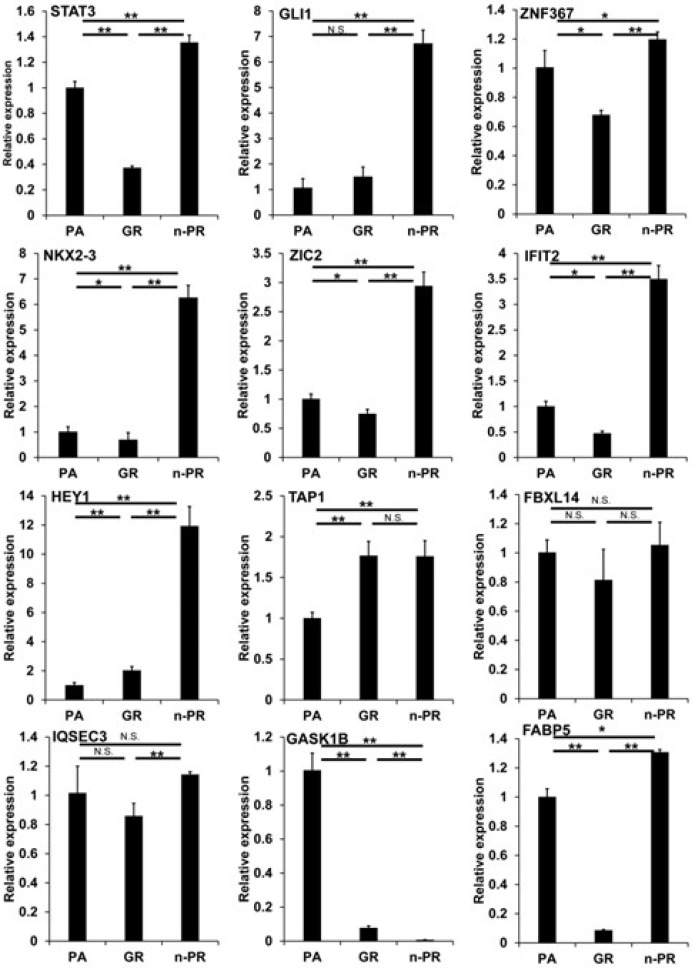
qPCR analysis of several candidate regulatory genes among 2D cultured MIA capa-2-PA, -GR and -PR. Among the 2D cultured cells obtained from 2D cultured MIA PaCa-2-PA, -GR and -PR, the mRNA expression of *STAT3*, *GLI1*, *ZNF367*, *NKX3-2*, *ZIC2*, *IFIT2*, *NEY1*, *TAP1*, *FBXL14*, *IQSEC3*, *GASK1B* and *FABP5* were evaluated by a qPCR procedure. All experiments were performed in triplicate, each of which involved the use of freshly prepared 2D structures (*n* = 3) in each experimental condition. * *p* < 0.05, ** *p* < 0.01, N.S. not significant.

**Table 1 biomedicines-12-01011-t001:** Significant up-regulated DEGs upon chemoresistance against GEM or PTX (Log2Fold change more than 6).

GEM Resistance				PTX Resistance			
Symbol	Gene Name	log2FoldChange	Adjusted *p* Value	Symbol	Gene Name	log2FoldChange	Adjusted *p* Value
GRIP1	Glutamate Receptor Interacting Protein 1	10.11002022	5.81494 × 10^−16^	ABCB1	ATP Binding Cassette Subfamily B Member 1	11.82920221	4.42378 × 10^−22^
ODAPH	Odontogenesis Associated Phosphoprotein	8.177849336	1.62602 × 10^−10^	VCAM1	Vascular Cell Adhesion Molecule 1	11.23331676	8.36206 × 10^−20^
IRF8	Interferon Regulatory Factor 8	8.096196874	6.19306 × 10^−10^	DACH1	Dachshund Family Transcription Factor 1	8.692296926	2.18033 × 10^−11^
APBA1	Amyloid Beta Precursor Protein Binding Family A Member 1	8.095610968	2.81002 × 10^−10^	PACRG	Parkin Coregulated	8.12989827	1.00025 × 10^−9^
A2M	Alpha-2-Macroglobulin	7.915319906	1.01049 × 10^−9^	PLCXD3	Phosphatidylinositol Specific Phospholipase C X Domain Containing 3	8.050077572	4.28566 × 10^−26^
DACH1	Dachshund Family Transcription Factor 1	7.897630617	2.35246 × 10^−9^	TMEM117	Transmembrane Protein 117	7.89400307	6.66321 × 10^−9^
VCAM1	Vascular Cell Adhesion Molecule 1	7.344145134	6.44933 × 10^−8^	BOC	BOC Cell Adhesion Associated, Oncogene Regulated	7.506336829	2.65139 × 10^−22^
ROBO2	Roundabout Guidance Receptor 2	7.334600056	6.92722 × 10^−8^	SERPINA2	Serpin Family A Member 2 (Gene/Pseudogene)	6.806740816	3.96974 × 10^−6^
TNFSF18	TNF Superfamily Member 18	7.318544705	2.44237 × 10^−16^	TNFSF8	TNF Superfamily Member 8	6.783054485	3.48679 × 10^−9^
SCN9A	Sodium Voltage-Gated Channel Alpha Subunit 9	6.867407737	5.16415 × 10^−14^	PRKN	Parkin RBR E3 Ubiquitin Protein Ligase	6.404876004	2.54818 × 10^−15^
MACC1	MET Transcriptional Regulator MACC1	6.827970039	1.36689 × 10^−6^	CYP26B1	Cytochrome P450 Family 26 Subfamily B Member 1	6.396521824	4.9334 × 10^−33^
PACRG	Parkin Coregulated	6.783843713	1.86292 × 10^−6^	HLA-DPA1	Major Histocompatibility Complex, Class II, DP Alpha 1	6.329226609	1.08798 × 10^−5^
PLOD2	Procollagen-Lysine,2-Oxoglutarate 5-Dioxygenase 2	6.690604956	7.77639 × 10^−7^	ADGRL2	Adhesion G Protein-Coupled Receptor L2	6.064776579	4.21876 × 10^−5^
UST	Uronyl 2-Sulfotransferase	6.298207932	1.29991 × 10^−11^	NCAM2	Neural Cell Adhesion Molecule 2	6.004166304	2.0329 × 10^−159^
MYRIP	Myosin VIIA And Rab Interacting Protein	6.28248296	2.89914 × 10^−5^				
NMUR1	Neuromedin U Receptor 1	6.144865541	1.18787 × 10^−5^				
ZDHHC15	Zinc Finger DHHC-Type Palmitoyltransferase 15	6.113699772	6.62112 × 10^−5^				

**Table 2 biomedicines-12-01011-t002:** Significant down-regulated DEGs upon chemoresistance against GEM or PTX (Log2Fold).

GEM Resistance				PTX Resistance			
Symbol	Gene Name	log2FoldChange	Adjusted *p* Value	Symbol	Gene Name	log2FoldChange	Adjusted *p* Value
TGFBR2	Transforming Growth Factor Beta Receptor 2	−11.03874963	3.32485 × 10^−19^	SCG2	Secretogranin II	−12.26545108	4.38044 × 10^−47^
DOC2B	Double C2 Domain Beta	−10.54986005	1.2234 × 10^−17^	DSG2	Desmoglein 2	−11.03564137	3.92949 × 10^−19^
VCAN	Versican	−9.882780644	2.8229 × 10^−131^	TGFBR2	Transforming Growth Factor Beta Receptor 2	−11.00861544	4.97415 × 10^−19^
SERPINA3	Serpin Family A Member 3	−9.550637947	2.83547 × 10^−14^	MX2	MX Dynamin Like GTPase 2	−10.67927343	7.7246 × 10^−18^
MDFIC	MyoD Family Inhibitor Domain Containing	−8.328703966	6.0121 × 10^−11^	CLDN11	Claudin 11	−10.56317073	7.44249 × 10^−35^
** FABP5 **	** Fatty Acid Binding Protein 5 **	** −8.128327037 **	** 2.08855 × 10^−20^ **	GNG2	G Protein Subunit Gamma 2	−10.26375148	2.25839 × 10^−16^
ATP2A3	ATPase Sarcoplasmic/Endoplasmic Reticulum Ca^2+^ Transporting 3	−8.122644241	4.78128 × 10^−10^	VCAN	Versican	−10.23159776	6.9482 × 10^−108^
LDB2	LIM Domain Binding 2	−8.01028135	9.87812 × 10^−10^	ARHGEF28	Rho Guanine Nucleotide Exchange Factor 28	−9.391001012	7.63866 × 10^−14^
LRP1B	LDL Receptor Related Protein 1B	−7.964026883	1.78675 × 10^−9^	EN2	Engrailed Homeobox 2	−9.049384354	2.29997 × 10^−12^
SLC6A12	Solute Carrier Family 6 Member 12	−7.784394113	9.93647 × 10^−25^	POPDC3	Popeye Domain Containing 3	−8.848875402	3.00989 × 10^−12^
GLIS3	GLIS Family Zinc Finger 3	−7.625226831	1.48895 × 10^−8^	MDFIC	MyoD Family Inhibitor Domain Containing	−8.297386286	1.03966 × 10^−10^
** IQSEC3 **	** IQ Motif And Sec7 Domain ArfGEF 3 **	** −7.585445172 **	** 1.34686 × 10^−8^ **	NRP2	Neuropilin 2	−8.047124286	1.57081 × 10^−9^
** FABPP7 **				LRP1B	LDL Receptor Related Protein 1B	−7.930614778	4.90813 × 10^−9^
TBC1D4	TBC1 Domain Family Member 4	−7.244628012	1.11093 × 10^−62^	AFAP1L2	Actin Filament Associated Protein 1 Like 2	−7.895166879	1.36679 × 10^−9^
SFMBT2	Scm-like With Four Mbt Domains 2	−7.17926521	1.22166 × 10^−15^	SERPINE2	Serpin Family E Member 2	−7.67590965	2.61181 × 10^−29^
OAS2	2’−5’-Oligoadenylate Synthetase 2	−7.175482694	1.38074 × 10^−10^	GLIS3	GLIS Family Zinc Finger 3	−7.592041864	4.36536 × 10^−8^
** IQSEQ3-AS1 **				PKP2	Plakophilin 2	−7.590189224	4.94123 × 10^−23^
IFI16	Interferon Gamma Inducible Protein 16	−7.130784594	4.52177 × 10^−30^	MMP1	Matrix Metallopeptidase 1	−7.566940209	4.88886 × 10^−8^
CSF3	Colony Stimulating Factor 3	−6.892382117	3.40616 × 10^−7^	CAMK2D	Calcium/Calmodulin Dependent Protein Kinase II Delta	−7.503432725	6.36716 × 10^−17^
AIM2	Absent In Melanoma 2	−6.886249925	2.64704 × 10^−7^	CPPED1	Calcineurin Like Phosphoesterase Domain Containing 1	−7.468614671	7.35512 × 10^−8^
IFI44	Interferon Induced Protein 44	−6.878989804	1.24045 × 10^−6^	DOC2B	Double C2 Domain Beta	−7.453740156	4.46885 × 10^−49^
NFATC2	Nuclear Factor Of Activated T Cells 2	−6.739446934	5.97491 × 10^−7^	TBX18	T-Box Transcription Factor 18	−7.290134659	2.18866 × 10^−7^
SLC2A8	Solute Carrier Family 2 Member 8	−6.648328287	3.30192 × 10^−6^	CALB2	Calbindin 2	−7.253641802	1.0662 × 10^−131^
DKK1	Dickkopf WNT Signaling Pathway Inhibitor 1	−6.642581097	9.2718 × 10^−77^	ZNF860	Zinc Finger Protein 860	−7.081806695	1.52466 × 10^−7^
MMP1	Matrix Metallopeptidase 1	−6.638877054	1.17851 × 10^−6^	COBL	Cordon-Bleu WH2 Repeat Protein	−7.048878829	5.36031 × 10^−10^
P2RY6	Pyrimidinergic Receptor P2Y6	−6.634829764	5.48177 × 10^−43^	TENT5A	Terminal Nucleotidyltransferase 5A	−6.889558144	2.11897 × 10^−58^
KRT16	Keratin 16	−6.625092523	4.61517 × 10^−6^	MSX2	Msh Homeobox 2	−6.817574756	3.37274 × 10^−6^
GARIN2	Golgi Associated RAB2 Interactor Family Member 2	−6.526874965	1.67813 × 10^−6^	IL1RAPL1	Interleukin 1 Receptor Accessory Protein Like 1	−6.656651415	7.79896 × 10^−6^
DSC2	Desmocollin 2	−6.491508652	8.7924 × 10^−6^	PTPRZ1	Protein Tyrosine Phosphatase Receptor Type Z1	−6.567655149	1.32301 × 10^−5^
NPR1	Natriuretic Peptide Receptor 1	−6.43952103	1.2944 × 10^−5^	** GASK1B **	** Golgi Associated Kinase 1B **	** −6.554441311 **	** 3.30097 × 10^−28^ **
ARHGDIB	Rho GDP Dissociation Inhibitor Beta	−6.396044956	1.31964 × 10^−5^	MAST4	Microtubule Associated Serine/Threonine Kinase Family Member 4	−6.551727171	1.51586 × 10^−5^
ARHGAP15	Rho GTPase Activating Protein 15	−6.371386344	1.49688 × 10^−5^	DSC2	Desmocollin 2	−6.458681971	2.63615 × 10^−5^
ZNF488	Zinc Finger Protein 488	−6.233311211	1.6178 × 10^−33^	PTGER4	Prostaglandin E Receptor 4	−6.432292978	3.3689 × 10^−5^
CX3CL1	C-X3-C Motif Chemokine Ligand 1	−6.232200898	3.1043 × 10^−5^	ARHGDIB	Rho GDP Dissociation Inhibitor Beta	−6.365076853	3.87778 × 10^−5^
MGP	Matrix Gla Protein	−6.096977355	6.32099 × 10^−5^	PTHLH	Parathyroid Hormone-like Hormone	−6.350109054	5.06751 × 10^−5^
STAC2	SH3 And Cysteine Rich Domain 2	−6.095588364	6.35438 × 10^−5^	ARHGAP15	Rho GTPase Activating Protein 15	−6.341010051	4.37253 × 10^−5^
RSAD2	Radical S-Adenosyl Methionine Domain Containing 2	−6.078148872	0.000133759	BLACAT1	BLACAT1 Overlapping LEMD1 Locus	−6.294580532	6.24095 × 10^−40^
STAMBPL1	STAM Binding Protein Like 1	−6.028954497	2.0585 × 10^−5^	JHY	Junctional Cadherin Complex Regulator	−6.290223158	6.09037 × 10^−5^
OASL	2’-5’-Oligoadenylate Synthetase Like	−6.027482168	5.31937 × 10^−44^	RIMKLB	Ribosomal Modification Protein RimK Like Family Member B	−6.257421445	1.60377 × 10^−52^
				ZNF488	Zinc Finger Protein 488	−6.200983103	1.60046 × 10^−31^
				PPP1R9A	Protein Phosphatase 1 Regulatory Subunit 9A	−6.15465423	1.70269 × 10^−20^
				NTS	Neurotensin	−6.132353346	3.6447 × 10^−7^
				MYO1B	Myosin IB	−6.072004859	0.000169145
				** GASK1B-AS1 **			
				KHDRBS3	KH RNA Binding Domain Containing, Signal Transduction Associated 3	−6.043385454	3.62909 × 10^−5^
				OXR1	Oxidation Resistance 1	−6.03633527	7.5147 × 10^−7^
				ENC1	Ectodermal-Neural Cortex 1	−6.019126253	2.6463 × 10^−100^

Bold and underlined DEGs are associated with their antisense or pseudogene.

**Table 3 biomedicines-12-01011-t003:** Possible upstream regulators and causal network master regulators.

	Upstream Regulator			Causal Network Regulator		Possible Role for Cancer Progression [Ref#]
PA vs GR	PA vs nPR	GR vs nPR	PA vs GR	PA vs nPR	GR vs nPR	
STAT3↓(Da)		STAT3↑(Sup)	STAT3↓(Da)			progressive [42]
	GLI1↑(Sur)	GLI1↑(Sur)		GLI1↑(Sur)	GLI1↑(Sur)	progressive [43]
	TP63↑(Da)	TP63↑(Da)				suppressive [44]
			TCF7↓(Da)	TCF7↓(Da)		progressive [45]
				ZNF367↓(Sur)	ZNF367↓(Sur)	suppressive [46]
				NKX3-2↑(Sur)	NKX3-2↑(Sur)	chemoresistant [47]
				ZNF703↓(Da)		progressive [48]
BHLHE↓(Da)						progressive [49]
	NFKBIA↑(Da)					suppressive [50]
					ZIC2↑(Sur)	progressive [51]
			WSL↓(Da)			progressive [52]
			IFIT2↓(Sur)			suppressive [53]
				SHOX2↓(Da)		progressive [54]
				HEY1↑(Sur)		radioresistance [55]
					FKBP4↓(Da)	progressive [56]
					CXCL8↓(Da)	chemoresistance [57]
					LTBR↓(Da)	progression [58]
					TAP1↑(Da)	progressive [59]
					FBXL14↓(Sur)	suppressive [60]

PA, parent; GR, gemcitabine resistant; PR, paclitaxel resistant; PA: Mia PaCa-2-PA; GR: MIA PaCa-2-GR; PR: MIA PaCa-2-PR; Da: cellular damage, Sur: survival (highlighted); ↑: up-regulation, ↓: down-regulation.

## Data Availability

The data that support the findings of this study are available from the corresponding author upon reasonable request.

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
