# Peer review of "Modulation of Epithelial–Mesenchymal Transition Is a Possible Underlying Mechanism for Inducing Chemoresistance in MIA PaCa-2 Cells against Gemcitabine and Paclitaxel"

_biomedicines, 2024, doi:10.3390/biomedicines12051011_

Round 1

Reviewer 1 Report

Comments and Suggestions for Authors

In this manuscript, the authors employed 2D and 3D cultures of parent MIA PaCa-2 cells (MIA PaCa-2-PA), their GEM resistance cell line (MIA PaCa-2-GR) and PTX resistance (MIA PaCa-2-PR) to study molecular mechanisms responsible for acquired resistance against GEM and PTX by using cellular metabolic and RNA sequencing analysis. Compared to the MIA PaCa-2-PA, MIA PaCa-2-GR or -PR showed a significant change in not only the formation of the 3D spheroids but also their mitochondrial and glycolytic functions. Such metabolic changes were also different between their 2D and 3D culture conditions. Furthermore, RNA sequencing data demonstrated that various modulatory factors related to EMT may be the critical regulators responsible for acquired drug resistance. Although the results are potentially interesting and helpful, there are several issues to be addressed before considering its acceptance.

Major comments:

1.     Compared to MIA PaCa-2-PA, metabolic changes of MIA PaCa-2-GR is different from those of PaCa-2-PR. The authors need to perform assays to further evaluate this discrepancy. If difficult to perform this assay, the authors should add the related explanation in the discussion section.

2.     Given significant changes of EMT factors are considered responsible for drug resistance following bioinformatical analysis, the authors should design the assays to further elucidate how these factors induce drug resistance. Otherwise, the authors should add further explanations in the discussion section.

Author Response

Dear Editor,

Thank you very much for the constructive comments concerning our manuscript, " Modulation of epithelial mesenchymal transition is a possible underlying mechanism for inducing chemoresistance in MIA PaCa-2 cells against gemcitabine and paclitaxel”. We carefully checked all of the Reviewer comments and prepared a revised version of our paper that takes these comments into account. The changes are listed below.

Reviewer 1

In this manuscript, the authors employed 2D and 3D cultures of parent MIA PaCa-2 cells (MIA PaCa-2-PA), their GEM resistance cell line (MIA PaCa-2-GR) and PTX resistance (MIA PaCa-2-PR) to study molecular mechanisms responsible for acquired resistance against GEM and PTX by using cellular metabolic and RNA sequencing analysis. Compared to the MIA PaCa-2-PA, MIA PaCa-2-GR or -PR showed a significant change in not only the formation of the 3D spheroids but also their mitochondrial and glycolytic functions. Such metabolic changes were also different between their 2D and 3D culture conditions. Furthermore, RNA sequencing data demonstrated that various modulatory factors related to EMT may be the critical regulators responsible for acquired drug resistance. Although the results are potentially interesting and helpful, there are several issues to be addressed before considering its acceptance.

Major comments:

  1. Compared to MIA PaCa-2-PA, metabolic changes of MIA PaCa-2-GR is different from those of PaCa-2-PR. The authors need to perform assays to further evaluate this discrepancy. If difficult to perform this assay, the authors should add the related explanation in the discussion section.

Answer; We sincerely appreciate your valuable comments. According to this suggestion, we have added the new paragraph regarding metabolic changes among the types of cells in the revised discussion section as followings; “The present study showed that metabolic capacities were significantly increased in MIA PaCa-2-GR and MIA PaCa-2-PR compared to those in MIA PaCa-2-PA. Such increased metabolic capacities were observed in both 2D- and 3D-culture conditions. Activation in metabolic pathways has been reported as one phenotype of metabolic plasticity, a finding that can be observed in cancer cells that are resistance to chemotherapy [75], which is consistent with the findings in the present study. The molecular mechanisms underlying chemotherapy resistance-induced metabolic alteration are presumably multifactorial, but the most plausible interpretation would be a compensatory response of cancer cells to secure energy for survival against anti-cancer agents. Indeed, it has also been reported that EMT can activate both oxidative phosphorylation and glycolysis [76]. Interestingly, the degree of increased metabolic capacity in MIA-PaCa-2-GR was milder than that in MIA-PaCa-2-PR. The finding that the gene expression level of FABP5, an important lipid chaperon for the activation of intracellular metabolism in cancer cells [77,78], was markedly different between MIA-PaCa-2-GR and MIA-PaCa-2-PR might be one of explanations for the differences in metabolic capacities between two cells. Nevertheless, an enhanced metabolic capacity in chemotherapy-resistant cells may play a significant role in cell survival and function. The assessment of cellular metabolism in the acquisition of chemotherapy resistance may be an important factor in the selection of anti-cancer therapies.”.

  1. Given significant changes of EMT factors are considered responsible for drug resistance following bioinformatical analysis, the authors should design the assays to further elucidate how these factors induce drug resistance. Otherwise, the authors should add further explanations in the discussion section.

Answer; We sincerely appreciate your valuable comments. According to this suggestion, we included additional information related to the possible role of VCAM1 which was identified as the top 7 and the top 2 significantly up-regulated DEG of GEM resistant and PTX resistant MIA Pa Ca2 cells, and in fact VCAM1 is already identified to be involved in the EMT, chemoresistance and cancer stemness in various cancer including pancreatic cancer. This information is included in the 1st paragraph in Discussion; “Furthermore, VCAM1 was identified as the top 7 and the top 2 significant upregulated DEG in MIA-GR and MIA-PR, respectively (Table 1). In fact, VCAM1 has been suggested as a factor to estimate poor patient prognosis and can promote tumor metastasis by inducing EMT in cancer [59]. In pancreatic cancer, EMT is known to lead acquiring the characteristics of cancer stemness and enhance chemotherapy resistance through multiple different ways [60]. For instance, it is reported that the recapitulation of the fibrotic rigidities in pancreatic cancer tissues promote elements of EMT, and the stiffness induces chemoresistance in pancreatic cancer cells [61]. Based on these findings, we suggested that VCAM1 is a potential regulator of the acquired resistance to GEM and n-PTX, which needs further investigation in the future.”.

Reviewer 2

The authors aimed to elucidate the currently unknown molecular mechanisms responsible for the similarity and difference during the acquirement of resistance against gemcitabine (GEM) and paclitaxel (PTX) in patients with pancreatic carcinoma by 3D spheroid configurations, mitochondrial metabolic functions, and RNA sequencing analysis. RNA sequencing and bioinformatic analyses of the differentially expressed genes (DEGs) suggested that various modulatory factors related to epithelial mesenchymal transition (EMT) including STAT3, GLI1, ZNF367, NKX3-2, ZIC2, IFIT2, HEY1 and FBLX, may be the possible upstream regulators and/or causal network master regulators responsible for the acquirement of drug resistance in MIA PaCa-2-GR and -PR. The results of this study were suggested that modulations of cancerous EMT may be key molecular mechanisms for inducing chemoresistance against GEM or PTX in MIA PaCa-2 cells. The topic of this study is interested. However, there are some issues to consider.

Major concern

  1. I suggest authors discuss the regulation of metabolic-related genes expressions in MIA PaCa-2-PA, MIA PaCa-21 2-GR, and MIA PaCa-2-PR. Whether the cancer cells exhibit a gain of function with chemoresistance by regulation of metabolic-related gene and protein expressions?

Answer; We sincerely appreciate your valuable comments. According to this suggestion, we have added the new paragraph regarding metabolic changes among the types of cells in the revised discussion section as followings; “The present study showed that metabolic capacities were significantly increased in MIA PaCa-2-GR and MIA PaCa-2-PR compared to those in MIA PaCa-2-PA. Such increased metabolic capacities were observed in both 2D- and 3D-culture conditions. Activation in metabolic pathways has been reported as one phenotype of metabolic plasticity, a finding that can be observed in cancer cells that are resistance to chemotherapy [75], which is consistent with the findings in the present study. The molecular mechanisms underlying chemotherapy resistance-induced metabolic alteration are presumably multifactorial, but the most plausible interpretation would be a compensatory response of cancer cells to secure energy for survival against anti-cancer agents. Indeed, it has also been reported that EMT can activate both oxidative phosphorylation and glycolysis [76]. Interestingly, the degree of increased metabolic capacity in MIA-PaCa-2-GR was milder than that in MIA-PaCa-2-PR. The finding that the gene expression level of FABP5, an important lipid chaperon for the activation of intracellular metabolism in cancer cells [77,78], was markedly different between MIA-PaCa-2-GR and MIA-PaCa-2-PR might be one of explanations for the differences in metabolic capacities between two cells. Nevertheless, an enhanced metabolic capacity in chemotherapy-resistant cells may play a significant role in cell survival and function. The assessment of cellular metabolism in the acquisition of chemotherapy resistance may be an important factor in the selection of anti-cancer therapies.”.

  1. Cancer stemness promotes cancer chemoresistance. Whether cancer stemness plays a role in the EMT-induced cancer chemoresistance in this study?

Answer; We sincerely appreciate your valuable comments. According to this suggestion, we included additional information related to the possible role of VCAM1 which was identified as the top 7 and the top 2 significantly up-regulated DEG of GEM resistant and PTX resistant MIA Pa Ca2 cells, and in fact VCAM1 is already identified to be involved in the EMT, chemoresistance and cancer stemness in various cancer including pancreatic cancer. This information is included in the 1st paragraph in Discussion; “Furthermore, VCAM1 was identified as the top 7 and the top 2 significant upregulated DEG in MIA-GR and MIA-PR, respectively (Table 1). In fact, VCAM1 has been suggested as a factor to estimate poor patient prognosis and can promote tumor metastasis by inducing EMT in cancer [59]. In pancreatic cancer, EMT is known to lead acquiring the characteristics of cancer stemness and enhance chemotherapy resistance through multiple different ways [60]. For instance, it is reported that the recapitulation of the fibrotic rigidities in pancreatic cancer tissues promote elements of EMT, and the stiffness induces chemoresistance in pancreatic cancer cells [61]. Based on these findings, we suggested that VCAM1 is a potential regulator of the acquired resistance to GEM and n-PTX, which needs further investigation in the future.”.

Minor concern

  1. Line 119, 120, 186 “@M” should be “mM”.

Answer; Thank you for this comment. As pointed out, those were fixed.

  1. Line 137, 143 “CO2” should be “CO2”.

Answer; Thank you for this comment. As pointed out, those were fixed.

  1. Comments on the Quality of English Language. Minor editing of English language required.

Answer; Thank you for this comment. As suggested, manuscript was edited by a native English speaking Scientist, Stewart Chisholm.

Reviewer 2 Report

Comments and Suggestions for Authors

The authors aimed to elucidate the currently unknown molecular mechanisms responsible for the similarity and difference during the acquirement of resistance against gemcitabine (GEM) and paclitaxel (PTX) in patients with pancreatic carcinoma by 3D spheroid configurations, mitochondrial metabolic functions, and RNA sequencing analysis. RNA sequencing and bioinformatic analyses of the differentially expressed genes (DEGs) suggested that various modulatory factors related to epithelial mesenchymal transition (EMT) including STAT3, GLI1, ZNF367, NKX3-2, ZIC2, IFIT2, HEY1 and FBLX, may be the possible upstream regulators and/or causal network master regulators responsible for the acquirement of drug resistance in MIA PaCa-2-GR and -PR. The results of this study were suggested that modulations of cancerous EMT may be key molecular mechanisms for inducing chemoresistance against GEM or PTX in MIA PaCa-2 cells. The topic of this study is interested. However, there are some issues to consider.

Major concern

l   I suggest authors discuss the regulation of metabolic-related genes expressions in MIA PaCa-2-PA, MIA PaCa-21 2-GR, and MIA PaCa-2-PR. Whether the cancer cells exhibit a gain of function with chemoresistance by regulation of metabolic-related gene and protein expressions?

l   Cancer stemness promotes cancer chemoresistance. Whether cancer stemness plays a role in the EMT-induced cancer chemoresistance in this study?

Minor concern

Line 119, 120, 186 “@M” should be “mM”.

Line 137, 143 “CO2” should be “CO2”.

Comments on the Quality of English Language

Minor editing of English language required

Author Response

(The authors gave the same response as above.)

Round 2

Reviewer 1 Report

Comments and Suggestions for Authors The authors are good enough to address all the issues mentioned in my review report. Therefore, this paper has been sufficiently improved to warrant publication in Biomedicines

Reviewer 2 Report

Comments and Suggestions for Authors

The authors have revised the manuscript according to the reviewer's comments. The manuscript is acceptable for publication.